# Assessing Genetic Variation among *Strychnos spinosa* Lam. Morphotypes Using Simple Sequence Repeat Markers

**DOI:** 10.3390/plants12152810

**Published:** 2023-07-28

**Authors:** Zoliswa Mbhele, Godfrey Elijah Zharare, Clemence Zimudzi, Nontuthuko Rosemary Ntuli

**Affiliations:** 1Department of Botany, Faculty of Science, Agriculture and Engineering, University of Zululand, KwaDlangezwa 3886, South Africa; mbhelez@unizulu.ac.za; 2Department of Agriculture, University of Zululand, KwaDlangezwa 3886, South Africa; zharareg@unizulu.ac.za; 3Department of Biological Sciences and Ecology, Faculty of Science, University of Zimbabwe, Harare P.O. Box MP167, Zimbabwe; czimudzi63@gmail.com

**Keywords:** *Strychnos spinosa*, genetic variation, domestication, crop improvement, food security

## Abstract

*Strychnos spinosa* Lam., commonly known as green monkey orange, is a highly valued indigenous fruit tree in South Africa with potential for domestication and commercialization. However, no study has reported on the molecular diversity of *Strychnos spinosa* morphotypes. Therefore, this study aimed to determine genetic variation among 32 *Strychnos spinosa* morphotypes using simple sequence repeat (SSR) markers. Fourteen amplified SSR markers produced 159 alleles, with a mean of 5.68 per locus. The polymorphic information content (PIC) values ranged from 0.22 (Ssp_1) to 0.84 (Ssp_6). Morphotypes were clustered in a biplot based on their genetic distances. The dendrogram chiefly discriminated morphotypes according to variation of pericarp texture. The population structure had the highest delta value K = 3, thus the 32 morphotypes were divided into three subpopulations based on the Bayesian approach. The affinities produced by the population structure agreed with the genetic distance of closely related morphotypes. This study is the first to report on SSR marker development and their successful use for genetic diversity and population structure studies of *Strychnos spinosa*. It provides insights into the molecular characterisation of *Strychnos spinosa*. This can lead to breeding programs and crop improvement programs, particularly in varietal developmental programs, which can contribute to alleviating food security challenges.

## 1. Introduction

*Strychnos spinosa* is one of the large-fruited *Strychnos* species native to sub-Saharan Africa, which has potential for domestication and commercialization for its edible fruit value [1]. Domestication of wild plants is complicated by the fact that the plants have high genetic diversity coupled with equally high morphological diversity between and within populations [2]. Studies reported by Mbhele et al. [3] have revealed distinct morphotypes based on the following traits: colour of recently sprouted but open leaves (young leaves), colour, shape, and form of fully developed leaves, as well as colour, texture, and shape of the immature fruits. It is therefore logical to conduct genetic analyses of the morphotypes to determine the underlying genetic characteristics of the morphotypes. Genetic studies of the morphotypes based on molecular markers would provide valuable information regarding the diversity, and would help to determine the relationship between the morphotypes at the molecular level. Genetic variation explains the evolutionary change or adaptive potential and dictates the phenotypic variation of any tree population or species, expressed across their morphological or physiological traits [4]. It provides raw materials for the variety of growth forms, yields, and wood qualities for production, leafing, and fruiting patterns, and adaptability to environmental changes and stresses, expressed differently by different tree species [5]. Generally, plant species with a diverse gene pool as in wild species and landraces, provide the genetic resources needed for the development of the plants for production in varied/diverse environments [6].

Genetic progress in characteristics desired for domestication is partly determined by the existing phenotypic diversity in the genetic base of plant populations [7]. Traditional breeding programs have always assessed intra-specific genetic and genomic variation within and among populations and families involving thousands of individuals in attempts to increase selection differential and, hence, maximize genetic and genomic gains [8]. An advantage of such a strategy is the control of environmental effects on the phenotype, which increases accuracy in computing breeding values and narrow-sense heritability estimates, provided the genotype x environment interaction and the number of genotypes that are evaluated are known [9]. Genetic diversity forms the basis of plant improvement and breeding [5].

All molecular marker systems that are developed and applied for analyses of genetic diversity and relatedness have their strengths and weaknesses [10]. Molecular markers have become an important instrument for characterizing wild and cultivated germplasm during the last decades [11]. Simple sequence repeat (SSR) markers are the most suitable genetic markers due to their multi-allelic nature and co-dominant inheritance, large genome coverage, small amount of starting DNA required, easy detection by polymerase chain reaction, and high polymorphism [5,12]. However, there are no microsatellite (SSR) markers available for *Strychnos spinosa,* and currently no study has been reported on the molecular diversity of *Strychnos spinosa* morphotypes. This study aimed to determine genetic diversity among *Strychnos spinosa* morphotypes using SSR markers, which is one of the prioritised tasks in a germplasm repository for fruit trees [13]. Determining the genetic basis of the morphological diversity among various *Strychnos spinosa* morphotypes identified by Mbhele et al. [3] will help identify genes for future breeding programs.

## 2. Results

In the current study, a total of fourteen simple sequence repeat primer pairs were successfully amplified for 32 *Strychnos spinosa* morphotypes (Table 1). The reliability was predicated on the distinct, constituent amplification of well-defined and expected alleles.

### 2.1. Genetic Variability among Strychnos spinosa Morphotypes

In *Strychnos spinosa* morphotypes, the allele size ranged from 140 bp (Ssp_13_F) to 407 bp (Ssp_18_F), with an average size of 270.57 bp (Table 1). The fourteen analysed SSR loci produced a total of 159 alleles, which ranged from two (Ssp_1_R) to twelve (Ssp_6_F), with a mean of 5.68 alleles per marker. The major allele frequency ranged from 0.24 (Ssp_7_R) to 0.85 (Ssp_1_R), with a mean of 0.51.

The forward marker Ssp_6 had the highest genetic diversity (GD = 0.82), whereas the reverse marker Ssp_7 had the lowest (GD = 0.24), with a mean of 0.62 (Table 1). The observed heterozygosity in *Strychnos spinosa* ranged from 0.00 (Ssp_13) to 0.88 (Ssp_11), with an average of 0.43, and the expected heterozygosity varied from 0.28 to 0.82 from loci Ssp_13, Ssp_6, and Ssp_11, respectively, with an average value of 0.60. The highest polymorphism (PIC = 0.84) was recorded in the forward marker Ssp_6, while the lowest (PIC = 0.22) was found in reverse marker Ssp_1.

### 2.2. Genetic Distance between Strychnos spinosa Morphotypes Based on Simple Sequence Repeat Markers

The genetic distance varied from 0.10 to 1.26 (Appendix A). Morphotype GSR-GRO had the closest genetic distance (GD = 0.10) to morphotype GRP-GEO. The genetic distance between morphotypes GRxCP-dGEF and GRP-dGEO was 0.20. Morphotypes GvRR-dGEO and GRR-GEF had a genetic distance of 0.30. Morphotypes PRR-dGEF and GRR-dGRO had the farthest genetic distance (GD = 1.26) with GRXCR-dGEF. The genetic distance between GRXCP-GEO and GRXCR-GEF; GRXCP-dGEF and GRXCR-dGEF; PRR-dGEF and GRR-dGRO; GRXCR-GEF and GRR-GEO, was also 1.26.

### 2.3. Population Structure among Strychnos spinosa

The Evanno test found a sharp strong maximum for Delta K at K = 3 in the plots of L (K) versus Delta (Figure 1). Thus, it clustered the *Strychnos spinosa* morphotypes into three sub-populations. The population structure grouped the genetic relationships of the morphotypes into sub-populations and admixtures as shown in K = 3 (Figure 2). The highest ∆K value was detected at K = 3 (Figure 1). The structure analysis clustered the 32 morphotypes into three sub-populations (K3.1 (Red), K3.2 (Green), and K3.3 (Blue)) at K = 3, based on their allele sizes. All the *Strychnos spinosa* morphotypes had admixtures. Morphotype GRXCR-dGEO as well as morphotypes GRP-GEO, GRP-dGEO, and GRXCP-GEF had K3.1 and K3.2 admixtures in K3 structure, but in opposite proportions. The remaining morphotypes had admixtures of K3.1, K3.2, and K3.3 sub-populations. Morphotypes GRXCR-GEF, GRP-GEO, GRP-dGEO, and GSR-GRO contained admixtures with about 99% of K3.2, but only 1% of K3.1. Morphotypes PRXCP-dGEO, GRP-dGRO, GRR-dGEO, GRR-GEF, and GvRR-dGEO had almost 95% of K3.3, where both K3.1 and K3.2 contributed only 5% to the admixture. 

### 2.4. Principal Coordinate Analysis

In the principal coordinate analysis (PCoA), *Strychnos spinosa* morphotypes were grouped based on the genotypic distance (Figure 3). The first two components of the principal coordinates accounted for 26.38% of the total variation. Most of the morphotypes were associated with the first quadrat and they formed two distinct sub-clusters. The first sub-cluster (upper portion of the quadrant) had morphotypes GvRR-dGRO, GRXCP-dGEF, PRR-dGRO, GRP-dGEF, and GRR-GEO. The second sub-cluster (lower portion of the quadrant) had the following morphotypes: GRP-dGRO, GRXCR-dGRO, GSR-dGRF, GRP-dGEO, GRXCR-dGEF, GSR-GRO, and GRR-dGRO. 

The second quadrant was defined by morphotypes GRP-GRO, GSR-GEO, PRR-dGRF, PRXCP-dGEO, and GRR-GEF (Figure 3). The first sub-cluster in the third quadrant was associated with morphotypes GSR-GEF, GSXCR-dGRF, GRXCP-GEO, and GRR-GRO, whereas the second sub-cluster had GRR-dGEO, GRP-GEF, GvRxCR-GEF, and GvRR-GRO. In the fourth quadrat, morphotypes were clustered together; GRXCR-dGEO, GvRR-dGEO, GRXCR-dGEF, and GRXCR-GEF formed a cluster, but PRR-dGEF and PRXCP-GEO were neither related to the cluster nor to each other.

### 2.5. The Phylogenetic Relationship among Strychnos spinosa Morphotypes

The dendrogram based on Euclidean distance classified the morphotypes into two major clusters and two sub-clusters (Figure 4). Cluster I was made up of morphotypes that had fruits with both smooth and rough pericarp. This cluster was further sub-divided into Cluster IA, which is primarily defined by fruits with rough pericarp and roundish shaped leaves, whereas Cluster IB contains fruits with smooth pericarp and elongated shaped leaves. Cluster II associated morphotypes with fruits that had rough and corrugated pericarp as well as green, elongated, and folded leaves. 

## 3. Discussion

The present study is the first to report on simple sequence repeat (SSR) marker development and their successful use for genetic diversity and population structure studies of *Strychnos spinosa*. Many species lack sequence data that would enable them to be positioned on the plant tree of life, and this limits our understanding of diversity and evolution among the species, despite significant sequencing initiatives such as the DNA barcoding effort led by the International Barcode of Life (iBOL) consortium and supported by the African Centre for DNA Barcoding in South Africa [14]. A lack of suitable molecular markers is a major hindrance to genomic and genetic studies of plants [15]. Any crop improvement program starts with the identification of variability among the genotypes [16]. Thus, the availability of microsatellite markers in crop species of interest is essential for conducting genetic studies and facilitating the crop improvement program [17], and *Strychnos spinosa* is one of the important potential crops that essentially requires microsatellite markers. Additionally, this study is the first to document the wide genetic variation of *Strychnos spinosa* within the same geographical area, which may indicate that KwaZulu-Natal can be the primary, secondary, or even tertiary gene pool and centre of diversity for *Strychnos spinosa*. Areas with high species diversity need urgent conservation measures to secure this germplasm and ensure food security for future generations [6]. Wild species normally contain a diverse gene source for new alleles and are ideal for plant breeding programs [18]. Genetic diversity is an important criterion to consider in prioritizing populations for conservation purposes [19]. If proper and stringent conservation practices and policies are not implemented, these resources will completely disappear [6].

### 3.1. Allelic Profile of Simple Sequence Repeats

A range in the allele size from 140 to 407 (Table 1) among the 32 studied *Strychnos spinosa* morphotypes was similar to a range from 140 to 550 among *Psidium* genotypes in New Delhi, which were developed from microsatellites-enriched libraries [20]. Although the range is similar, the primers used for these species were different. The fourteen SSR markers detected a lesser total number of alleles (159) and a range of alleles per marker (2–12) among *Strychnos spinosa* morphotypes (Table 1) compared with the total number (207) and the range of alleles per marker (6–17) detected in *Passiflora edulis* Sims accessions [21]. A range in major allele frequency from 0.24 to 0.85 with an average of 0.51 in this study (Table 1) was similar to the range (0.17–0.94) and average (0.56) among *Psidium* genotypes [21].

The SSR allelic profiles revealed high levels of polymorphism (Table 1) and therefore have a great ability to discriminate closely related morphotypes [22]. Marker Ssp_6 F had the highest number of alleles and genetic diversity, which suggests that this SSR marker had the highest degree of polymorphism [23]. However, because of the different instruments, software, and genotypes used for SSR analysis, the detected allelic variations in terms of number, size, and major allele frequencies may differ slightly among various studies of the same species [22].

### 3.2. Genetic Diversity, Observed and Expected Heterozygosity, and Polymorphic Information Content

A range (0.26–0.86) and average (0.62) of genetic diversity among *Strychnos spinosa* morphotypes (Table 1) was greater than a range (0.11–0.88) and average (0.53) among *Psidium guajava* [20]. The average values of observed heterozygosity (0.43) expected heterozygosity (0.60) and polymorphic information content (0.57) in the current study (Table 1) were higher than those obtained in *Rubus* cultivars (H_o_ = 0.29, H_e_ = 0.36, PIC = 0.33) [23], *Pisium sativum* (H_o_ = 0.085, H_e_ = 0.170, PIC = 0.323) [24], and *Solanum lycopersicum* (H_o_ = 0.29, H_e_ = 0.36, PIC = 0.33) [25]. The higher level of expected heterozygosity than observed both on average and in most individual loci of the present study (Table 1) was an indication that the morphotypes were formed by a mixture of genotypes with different genetic backgrounds [20], in addition to indicating high genetic diversity [26]. It is normally expected to find an excess of expected heterozygosity in comparison with observed heterozygosity due to the Wahlund effect [27]. This effect is both the apparent excess of homozygotes and the deficit of heterozygotes, which occur at single loci in a large sample of individuals due to the existence of population subdivision. It occurs when individuals from different subpopulations with diverse allele frequencies are combined in a single sample [28]. The high expected heterozygosity suggests the possibility of an ongoing hybridization process among the morphotypes, as there is a wide range of diversity between the morphotypes, which might have resulted from perhaps genetic mutations, polyploidy, or cross pollination [29]. There is also a possibility for long-distance cross-pollination between *Strychnos spinosa* morphotypes and its relative *Strychnos cocculoides*, with whom they share similar phenotypic, sensory traits, and the same common name in Zimbabwe [30]. The polymorphic information content (PIC) values of the markers can provide an estimate of discrimination power in a set of accessions by taking into consideration both the number of alleles and the relative frequencies of each allele [16]. This indicates that the SSR markers will become a useful tool for genetic variation studies and for genotype identification and similarity analysis in *Strychnos spinosa*, which are properties required for plant breeding programs [18]. 

### 3.3. Genetic Distance

Genetic distance is used to determine the relatedness between morphotypes in a population [31]. In this study, correlations based on genetic distance ranging from 0.20 to 1.26 indicated that there is applicable genetic divergence among the 32 *Strychnos spinosa* morphotypes (Appendix A). Morphotypes GSR-GRO, GRP-GEO, GRxCP-dGEF, GRP-dGEO, GvRR-dGEO, and GRR-GEF were genetically the closest, and they also shared most of the phenotypic characteristics such as fruit colour, rough skin texture, and elongated leaves, which suggests that each of these traits is influenced by the same genes among the morphotypes [26] and perhaps may have arisen by hybridisation [32]. Future research on other *Strychnos spinosa* in different geographical regions can shed light on the genetic diversity and population structure of the species. Genetic variations reflect the viability and evolutionary potential of a natural population, which is crucial for understanding the evolutionary history of extant populations [33]. Existing levels of genetic diversity in plant populations vary depending on mating patterns, population density, and the consistency of geographical distribution, among other evolutionary scenarios [34]. The genetic diversity of natural populations must be preserved if species are to survive and evolve [33]. Therefore, the preservation of intra-population genetic diversity ought to be given top priority because ensuring the long-term persistence of species is the ultimate objective of conservation efforts [35]. 

Morphotypes PRR-dGEF and GRR-dGRO had the farthest genetic distance (GD = 1.26) with GRXCR-dGEF. The genetic distance between GRXCP-GEO and GRXCR-GEF; GRXCP-dGEF and GRXCR-dGEF; PRR-dGEF and GRR-dGRO; and GRXCR-GEF and GRR-GEO, was also 1.26. This suggests that these morphotypes may be potentially good candidates for breeding. Breeding genetically distant morphotypes can result in a heterotic effect [36]. Heterosis plays a crucial role in breeding programs and leads to remarkable improvements in yield, quality, and earliness, which are considered the desirable outcomes when genetically distant parents are used to produce hybrid off-springs [37].

### 3.4. Population Structure

The population structure for K = 3 (Figure 2) and the highest delta value that occurred at K = 3 (Figure 1) indicated that the morphotypes can be divided into three subpopulations with admixed morphotypes amongst the subpopulations. This demonstrates a complex history of gene flow and admixtures among the different populations [38], which may have contributed to the observed morphological diversity of *Strychnos spinosa* [3]. The affinities produced by population structure (Figure 2) generally agreed with the genetic distance (Appendix A) of closely related morphotypes GSR-GRO and GRP-GEO in sub-population K3.2, and closely related morphotypes GRR-GEF and GvRR-dGEO in sub-population K3.3. In crop breeding, knowing the population structure benefits the selection of parents with genetic divergence [39]. Knowledge of the population structure can also aid in identifying regions of the genome that are under selection, which can provide insight into the adaptation of the crop to its environment [40], which is essential for plant conservation. This information can also be used to develop breeding strategies for improving crop yield and resilience to changing environmental conditions [6], as *Strychnos spinosa* is resilient to unfavourable conditions.

### 3.5. Principal Coordinate Analysis and Phylogenetic Relationship

Morphotypes GSR-GRO, GRP-GEO, GRXCP-dGEF, and GRP-dGEO, with narrow Nei’s genetic distance among each other (Appendix A) were grouped into one cluster of principal coordinate analysis (Figure 3) and in Cluster I of the dendrogram (Figure 4). Apparently, the closely related morphotypes GSR-GRO and GRP-GEO as well as morphotypes GRXCP-dGEF and GRP-dGEO, were each closely associated in Clusters IA and IB, respectively (Figure 4). This highest degree of similarity was probably due to their similarities in fruit colour, pericarp texture, fruit shape, and leaf colour, form, and shape. Therefore, the PCoA (Figure 3) and phylogenetic relationship (Figure 4) obtained from the microsatellite-based analyses grouped the closely related morphotypes as also established by genetic distance (Appendix A). These findings suggest that there is a strong correlation between genetic distance and morphological similarity among these morphotypes based on microsatellite analysis. Studies using additional genetic markers may also provide more insight into the evolutionary relationships among these morphotypes and their potential ecological roles in their respective habitats, because the ecological consequences of intraspecific diversity can also be investigated through the lens of variation of genotypic attributes [41].

The grouping of morphotypes in the dendrogram according to the variation of pericarp texture, using SSR markers (Figure 4) confirms the similar grouping based on the morphological traits [3]. The exceptions where the morphotypes were previously grouped on their own using the morphological traits [3] but are associated with the others in the SSR analysis (Figure 4), would indicate that those morphotypes were expressing different phenotypic traits that were from the same genomic origin. This probably explains the fading of roughness in the pericarp texture for some morphotypes during fruit growth, which requires further investigation. 

## 4. Materials and Methods

### 4.1. Plant Material and DNA Isolation

A representative set of *Strychnos spinosa* plant material was collected across the study area at Bonamanzi Game Reserve in KwaZulu-Natal to assess the genetic variation among the 32 morphotypes identified in the area (Table 2). Morphotype names were coined based on colour, texture, and shape of the immature fruits, colour of recently sprouted but open leaves (young leaves), as well as colour, shape, and form of fully developed leaves [3]. A minimum of 50 young (newly resprouted) leaves per plant were collected separately from three plants of each *Strychnos spinosa* morphotype, where each plant represented a replicate. Leaves were collected and immediately stored in silica gel for their transport to the laboratory. These leaves were then kept in a −80 °C freezer, and thereafter freeze-dried for 24 h. The DNA was extracted using the DNeasy Plant Mini Kit (QIAGEN^®^, Valencia, CA, USA) according to the manufacturer’s instructions. Freeze-dried leaves were ground using a mortar and pestle. For every plant per morphotype, 20 mg of powdered leaf tissue was added to 1.5 µL Eppendorf Safe-Lock Tubes (microcentrifuge tubes) together with 400 µL of Buffer AP1 and 4 µL of a 100 mg/mL RNase A stock solution. This mixture was thoroughly vortexed to eliminate any tissue clumps that may have formed. In order to lyse the cells, the mixture was incubated for 10 min in a 65 °C preheated water bath, with mixing two or three times during incubation by inverting tubes.

The detergent, proteins, and polysaccharides were co-precipitated by adding 130 µL Buffer AP2 to the lysate, mixed, and incubated for 5 min in ice. The lysate was centrifuged (using Eppendorf Mini Spin plus) for 5 min at 14,000 rpm, and the supernatant transferred (by pipette) into the QIAshredder Mini spin column placed in a 2 mL collection tube and centrifuged for 2 min at 14,000 rpm. This was performed to eliminate the effect of the DNA shearing that can result from the extreme viscous lysate and large amounts of precipitates that can be generated during this step.

The flow-through fraction was transferred to a new 1.5 mL microcentrifuge tube, without disturbing the cell-debris pellet. Usually, 450 µL of the lysate is recovered; therefore, 1.5 volumes (675 µL) of buffer AP3/E were added to the cleared lysate and immediately mixed by pipetting. During this stage, precipitation was produced. The 650 µL of the mixture, including any formed precipitate, was transferred using a pipette into the DNeasy Mini spin column in a 2 mL collection tube and centrifuged for 1 min at 8000 rpm, and the flow-through was discarded. The collecting tube was re-used, and this step was repeated with the remaining sample.

The DNeasy Mini spin column was placed into a new 2 mL collection tube, and 500 µL of Buffer AW was added and centrifuged for 1 min at 6000× *g* (8000 rpm). The flow-through was discarded while the collection tube was re-used. An additional 500 µL of Buffer AW was added and centrifuged at 14,000 rpm to dry the membrane. The DNeasy Mini spin column was transferred to a 1.5 mL or 2 mL microcentrifuge and 100 µL of Buffer AE was added onto the DNeasy membrane. This was incubated for 5 min at ambient temperature (15–25 °C), after which it was centrifuged for 1 min at 8000 rpm to elute. This process was performed in a separate microcentrifuge tube in order to obtain two identical sets of DNA with the same purity but differing concentrations.

### 4.2. Genotyping Using Simple Sequence Repeat Markers

The polymerase chain reaction (PCR) amplifications were performed by the Eppendorf mastercycler^®^ in 50 ng/µL of DNA template in two separate 10 µL volume reactions. The reactions contained 4 µL of DNA template, 0.8 µL of deoxyribonucleotide triphosphate (dNTPs) (2.5 mM), 1.0 µL of 10 × buffer and 0.06 µL of Taq polymerase (Inqaba Biotec, Pretoria, South Africa). In the first reaction, 1.0 µL of MgCl_2_ (50 mM), 1.0 µL of forward and reverse primers (5 µM), and 1.14 µL of ultrapure water were included. In the second reaction, a 1.2 µL of MgCl_2_ (50 mM) and 1.5 µL of both forward and reverse primers were added to make up the master mix. Forward primers were labelled with FAM (blue), ATTO565 (red), and ATTO550 (yellow), ATTO532 (green) fluorescent dyes. The PCR conditions consisted of denaturing at 94 °C for 2 min, nine cycles at 93 °C for 15 s, annealing at 65 °C for 20 s, and the extension at 72 °C for 30 s. The annealing temperature of each cycle was decreased by 1 °C with the final 30 cycles at 55 °C and the final elongation step at 72 °C for 5 min. The PCR products were separated by capillary electrophoresis analysis performed on an ABI3500 genetic analyser. Allele size was determined for each SSR locus using GeneMarker HID version 2.9.5. Of all primers tested, fourteen produced constituent amplification of well-defined allele sizes and were selected for further analysis (Table 1). 

### 4.3. Simple Sequence Repeats Analysis

Genetic diversity parameters, namely allele number and frequency, gene diversity, heterozygosity, and polymorphic information content (PIC), were calculated in PowerMarker software version 3.25. However, observed and expected heterozygosity were determined using Cervus version 3.0.7, where both the forward and reverse primers had one value for each marker. To clarify the gene differentiation between morphotypes, Nei’s genetic distance was evaluated. Values < 0.50 indicated closely related individuals, whereas ≥0.50 indicated distantly related morphotypes. The population structure analysis was determined using a Bayesian model-based clustering approach. The STRUCTURE version 2.3.4 program was applied to detect population genetic structure using a defined number of pre-set populations K, where each K is characterized by a set of allele frequencies at each locus. The Evanno test is recommended to identify the best-fitting number of populations within a sample. The structure program was set as follows: the analysis was run with 10 simulations per K value from K = 1 to 10, using a burn-in period length of 5000 and after burn-in 50,000 replicates. The most expected value of K for each test was detected by ΔK [42] using the Structure Harvester [43], online (http://tayloro.biology.ucla.edu/struct_harvest/, accessed on 20 October 2022). Bar plots were generated with mean results of runs for the highest K value using STRUCTURE version 2.3.4. The principal coordinate analysis (PCoA) was performed using GenAlEx version 6.4 software. The dendrogram was obtained using Ward’s method of linking based on Euclidean distance in XLSTAT version 2022.1.2 and displayed genetic relations among the *Strychnos spinosa* morphotypes.

## 5. Conclusions

This study provides insights into the molecular characterisation of *Strychnos spinosa* morphotypes. This paves the way for further molecular genetic investigations, further breeding programs, and crop improvement programs, particularly in the varietal developmental programs, which have the potential to contribute to alleviating food security challenges. Additionally, the findings of this study can also be used to inform conservation efforts and make informed conservation decisions for *Strychnos spinosa*, as well as contribute to the understanding of the evolutionary history and genetic diversity of this species in the region. This is the first study to report on genetic variation of *Strychnos spinosa.* Hence, this study was limited by the number of markers used (14 markers). Therefore, it is recommended that more markers be used in future studies.

## Figures and Tables

**Figure 1 plants-12-02810-f001:**
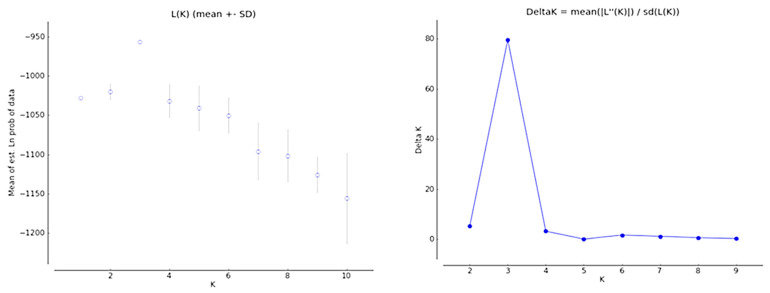
The Evanno test showing plot parameters of L (K) and Delta against the likely subpopulations of the 32 morphotypes.

**Figure 2 plants-12-02810-f002:**
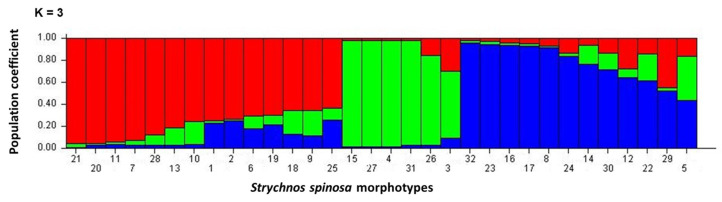
Population structure of 32 *Strychnos spinosa* morphotypes revealed by simple sequence repeat analysis K = 3; K3.1 (Red), K3.2 (Green), K3.3 (Blue). Landraces: 1, GRXCP-GEO; 2, GRR-GEO; 3, GSXCR-dGRF; 4, GRXCP-GEF; 5, GRP-GRO; 6, GvRR-dGRO; 7, GRXCP-dGEF; 8, PRXCP-dGEO; 9, PRR-dGRO; 10, GSR-GEF; 11, GvRR-GRO, 12, PRR-dGEF; 13, GRP-GEF; 14, GSR-dGRF; 15, GRP-GEO; 16, GSR-GEO; 17, GRP-dGRO; 18, GRR-dGEO; 19, GRR-dGRO; 20, GRR-GRO; 21, GRxCR-dGEO; 22, GRP-dGEF; 23, GRR-GEF; 24, PRR-dGRF; 25, GvRxCR-GEF; 26, PRXCP-GEO; 27, GRP-dGEO; 28, GRXCR-dGRO; 29, GRXCR-dGEF; 30, GRXCR-GEF; 31, GSR-GRO; 32, GvRR-dGEO, morphotypes are explained in Table 2.

**Figure 3 plants-12-02810-f003:**
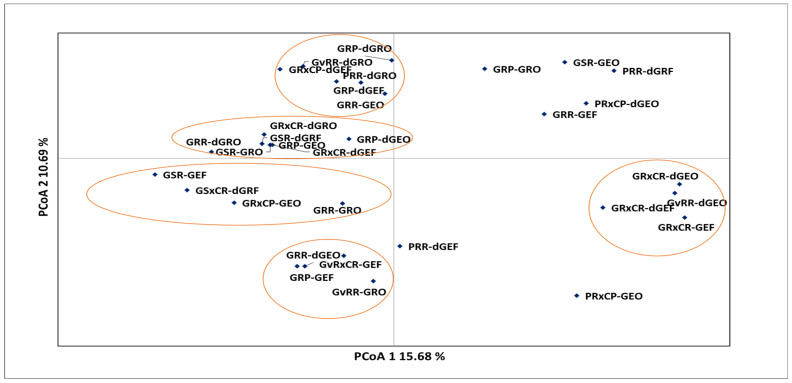
Principal coordinate analysis of *Strychnos spinosa* morphotypes from simple sequence repeat markers based on the genotypic distance. Morphotypes are explained in Table 2.

**Figure 4 plants-12-02810-f004:**
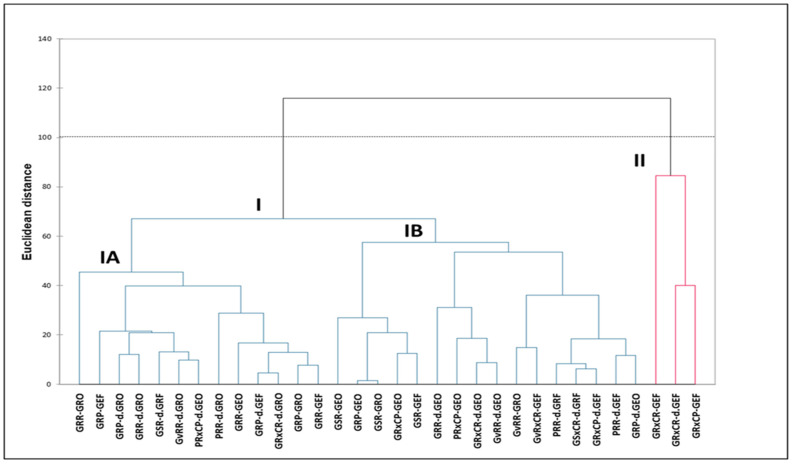
Phylogenetic relationship among *Strychnos spinosa*. Morphotypes are explained in Table 2.

**Table 1 plants-12-02810-t001:** Genetic variability among *Strychnos spinosa* morphotypes for fourteen simple sequence repeat markers.

Marker	Dye Used	Primer Sequences (5′-3′)	SSR Sequence	AS ^c^	S ^d^	AN ^e^	MAF ^f^	GD ^g^	H_0_ ^h^	H_e_ ^i^	PIC ^j^
Ssp_1_F ^a^	FAM	TGATGCAATGGATGTGTGCTAT	(ATTT)^6	144	32	4	0.76	0.40	0.36	0.35	0.39
Ssp_1_R ^b^	FAM	TGAAGACGGCAATGCGAACC			32	2	0.85	0.26			0.22
Ssp_2_F	ATTO532	TCGGAATACTACGGGCCACC	(AAAT)^5	199	32	4	0.58	0.58	0.31	0.34	0.52
Ssp_2_R	ATTO532	TCCCTTCCAACCCTTCAATAAC			32	4	0.61	0.55			0.49
Ssp_6_F	FAM	GCCAGACAAGTTTCCCTCGG	(ATTT)^6	239	32	12	0.27	0.86	0.73	0.82	0.84
Ssp_6_R	FAM	CCCGCGCTCAATGCTCTTAC			32	9	0.52	0.69			0.67
Ssp_7_F	FAM	TCTTTGCTTTCTTCCTCGAAAGG	(ATTT)^5	281	32	6	0.27	0.78	0.26	0.76	0.74
Ssp_7_R	FAM	GTATGATAGGTTCCACACGGC			32	5	0.24	0.77			0.74
Ssp_8_F	ATTO532	GCCTATGGCAAGCAATGTATTC	(AACT)^7	285	32	4	0.45	0.63	0.49	0.72	0.55
Ssp_8_R	ATTO532	CCTTGAGTTCCAAGCTGCAC			32	7	0.42	0.75			0.72
Ssp_9_F	ATTO550	CTGGACTGTCTTCTCGGGTTC	(AAAT)^5	288	32	6	0.76	0.41	0.50	0.51	0.40
Ssp_9_R	ATTO550	CAATTGCCAGTAACCGTGTAGG			32	4	0.52	0.55			0.46
Ssp_10_F	ATTO550	GACATACAAATAGAAGCACTGG	(ATTT)^5	181	32	5	0.70	0.49	0.34	0.59	0.46
Ssp_10_R	ATTO550	CATGAGGGAAACCCACCCTG			32	7	0.48	0.68			0.64
Ssp_11_F	ATTO565	ATTCTGGTCCCGTCACTGCC	(ATGC)^5	314	32	6	0.45	0.66	0.88	0.82	0.61
Ssp_11_R	ATTO565	CTTCGGGTGCCAAAGTTCAC			32	7	0.30	0.78			0.75
Ssp_12_F ^a^	FAM	TGCCTACTAACTAGCGTGAGG	(AAAT)^7	355	32	7	0.27	0.75	0.24	0.73	0.71
Ssp_12_R ^b^	FAM	AGCCAGCGAATTGTGTTATCC			32	9	0.33	0.74			0.70
Ssp_13_F	ATTO550	TCTATGTTGGAAATGCGCACG	(AATT)^5	140	32	4	0.64	0.53	0.00	0.28	0.47
Ssp_13_R	ATTO550	CATTGCACACAAAGCTACCTG			32	4	0.64	0.53			0.47
Ssp_14_F	ATTO550	GTTGGGGGTTAAACATTCAGC	(ATTT)^5	248	32	5	0.30	0.57	0.07	0.43	0.51
Ssp_14_R	ATTO550	CACTTTTATGCTCCCGTGTCC			32	4	0.33	0.55			0.48
Ssp_15_F	FAM	CAAGGTTTCGCCGAGCTGC	(AAAT)^6	377	32	5	0.36	0.72	0.29	0.73	0.68
Ssp_15_R	FAM	CTTGGAGTCCCAAGAAGCCG			32	5	0.36	0.74			0.69
Ssp_18_F	ATTO565	CAAAGCCCGAGGCATCAACC	(AAAT)^5	407	32	9	0.27	0.82	0.81	0.81	0.79
Ssp_18_R	ATTO565	GAAACCTGGTACGGGCAGC			32	6	0.42	0.71			0.67
Ssp_19_F	ATTO532	GATGGAGAGCCCAATGCAAG	(ATTT)^6	330	32	4	0.60	0.52	0.47	0.44	0.44
Ssp_19_R	ATTO532	GCTGTGAATTGTTAAAGGTCAAC			32	5	0.82	0.32			0.30
Mean				270.57	32	5.68	0.51	0.62	0.43	0.60	0.57

^a^ F—forward marker; ^b^ R—the reverse marker; ^c^ AS—allele size; ^d^ S—sample size; ^e^ AN—allele number; ^f^ MAF—major allele frequency; ^g^ GD—genetic diversity; ^h^ H_0_—observed heterozygosity; ^i^ H_e_—expected heterozygosity; ^j^ PIC—polymorphic information content.

**Table 2 plants-12-02810-t002:** Immature fruit and mature leaf attributes used to name *Strychnos spinosa* morphotypes [3].

Morphotype	Fruit Colour	Fruit Texture	Fruit Shape	Fully Grown Leaf Colour	Leaf Shape	Leaf Form
GRP-dGEF	Green	Rough	Pyriform	Dark green	Elongated	Folded
GRP-GEF	Green	Rough	Pyriform	Green	Elongated	Folded
GRP-dGEO	Green	Rough	Pyriform	Dark green	Elongated	Open
GRP-GEO	Green	Rough	Pyriform	Green	Elongated	Open
GRP-dGRO	Green	Rough	Pyriform	Dark green	Roundish	Open
GRP-GRO	Green	Rough	Pyriform	Green	Roundish	Open
GRR-dGEO	Green	Rough	Roundish	Dark green	Elongated	Open
GRR-GEO	Green	Rough	Roundish	Green	Elongated	Open
GRR-dGRO	Green	Rough	Roundish	Dark green	Roundish	Open
GRR-GRO	Green	Rough	Roundish	Green	Roundish	Open
GRR-GEF	Green	Rough	Roundish	Green	Elongated	Folded
GRxCP-dGEF	Green	Rough and corrugated	Pyriform	Dark green	Elongated	Folded
GRxCP-GEF	Green	Rough and corrugated	Pyriform	Green	Elongated	Folded
GRxCP-GEO	Green	Rough and corrugated	Pyriform	Green	Elongated	Open
GRxCR-dGEF	Green	Rough and corrugated	Roundish	Dark green	Elongated	Folded
GRxCR-GEF	Green	Rough and corrugated	Roundish	Green	Elongated	Folded
GRxCR-dGEO	Green	Rough and corrugated	Roundish	Dark green	Elongated	Open
GRxCR-dGRO	Green	Rough and corrugated	Roundish	Dark green	Roundish	Open
GSR-dGRF	Green	Smooth	Roundish	Dark green	Roundish	Folded
GSR-GEF	Green	Smooth	Roundish	Green	Elongated	Folded
GSR-GEO	Green	Smooth	Roundish	Green	Elongated	Open
GSR-GRO	Green	Smooth	Roundish	Green	Roundish	Open
GSxCR-dGRF	Green	Smooth and corrugated	Roundish	Dark green	Roundish	Folded
GvRR-dGEO	Green	Very rough	Roundish	Dark green	Elongated	Open
GvRR-dGRO	Green	Very rough	Roundish	Dark green	Roundish	Open
GvRR-GRO	Green	Very rough	Roundish	Green	Roundish	Open
GvRxCR-GEF	Green	Very rough	Roundish	Green	Elongated	Folded
PRR-dGRF	Purple	Rough	Roundish	Dark green	Roundish	Folded
PRR-dGEF	Purple	Rough	Roundish	Dark green	Elongated	Folded
PRR-dGRO	Purple	Rough	Roundish	Dark green	Roundish	Open
PRxCP-dGEO	Purple	Rough	Pyriform	Dark green	Elongated	Open
PRxCP-GEO	Purple	Rough	Pyriform	Green	Elongated	Open

## Data Availability

The data will be available from authors on request.

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
