# Peer review of "Assessing Genetic Variation among *Strychnos spinosa* Lam. Morphotypes Using Simple Sequence Repeat Markers"

_plants, 2023, doi:10.3390/plants12152810_

Round 1
Reviewer 1 Report
The article entitled: Assessing Genetic Variation Among Strychnos spinosa Lam. 2 Morphotypes using SSR markers, presents an analysis of 14 molecular markers of the SSR type in 32 morphotypes of Strychnos spinosa. The article is correctly stated, the introduction really frames the available knowledge and the results section clearly states the result obtained by using this type of molecular markers.
For this reviewer, the only question remains is related to how the primers used were developed, whether they were developed from some isolation strategy of the species of interest or were taken from previous reports in an interspecific manner, in which case it would be desirable to provide evidence of the sequence of the amplicons to be sure that a microsatellite region is being amplified. The discussion refers to previous work in Guava but does not specify the origin of the primers used.
The recommendation of this reviewer is to accept the article after having completed the information of the origin of the primers. The article presents an advance in the knowledge of genetics and population studies of this analyzed species.
Reviewer 2 Report
Comments and Suggestions for Authors
Dear Author,
It is my pleasure to review the manuscript entitled “Assessing Genetic Variation Among Strychnos spinosa Lam. Morphotypes using SSR markers” a research article submitted to MDPI Journal, Plants. Authors of this manuscript studied molecular diversity of Strychnos spinosa using 32 morphotypes and 14 SSR markers. They have performed series of genetic diversity related morpho-physiology study to select genotypes to be further used in breeding program for sustainable development. Overall, the experiments, they performed, are well and the results are convincing. Thus, the presented results take up an important topic consistent with the profile of the Journal. However, I have some suggestions, which might improve the manuscript to make important to the wider readers. Few suggestions I have mentioned in the main text pdf file. Please check.
-There are many places where grammar can be improved. I suggest a careful revision by an expert.
- Aauthors should elaborate and clarify the firm aim of the study precisely and simultaneously. Why diversity, genetic variation analysis is necessary of these materials?
Abstract: - Use common name also of the material for easy understanding
1. Introduction
-The Introduction also should be focused on the studied species even some far related research as available in the literature
-Also, it appears that the main aim of the study is to assess the species genetic diversity for conservation and breeding strategies and should be clearly stated and justified by relevant references.
-Introduction should reflect a little results summary. Rationale to be elucidated for the purpose of the study.
2. Results
-The values of observed heterozygosity is very high 0.88 (avg 0.62). Hence the scoring was erroneous and further analyses are less important.
-Forward and revers primers come together in PCR, why individual result for forward and reverse markers are presented?
-Table 2 is less important. You may use as supplement
-Fig. 1. Resolution must be improved
-The highest ∆K value was detected at K = 3 (Figure 1). Thus, population should be classified in to 3 sub-population. However, what does K=4 indicate? Is it necessary? If so, should be indicated clearly in the result.
3. Discussion
- Discussion is too lengthy. Many redundancies of statement and result presentation from Introduction, result and table.
-The authors should discuss more thoroughly the results obtained regarding further use of the morphotypes in breeding strategies for varietal improvement rather irrelevant discussion like marker development.
- Discuss whether the objectives of the study were reached and how the data can be further used in breeding
4. Materials and Methods
-In introduction you discussed the simplicity of SSR marker, however, in methods, DNA isolation process is very complex. Simple DNA isolation protocol is very available. I suggest you may test simple protocol to figure out differences and suggest to readers.
-Why table 4 is presented while Table 1 represents most information? It is just redundant
-Do you think simple PCR and agarose gel electrophoresis will give same result as you found from capillary electrophoresis analysis performed on an ABI3500 genetic analyzer which is very complex and expensive?
-More details on trials should be given such as inclusion of check varieties, how many plants/fruits/leaves were samples for each morphotype and what was the process?
-Similarly, details on statistical analysis of morphotyping is limited.
-For SSR analysis, only 14 markers were used. Are these linked to some traits/ or random markers were taken? The markers should be arranged chromosome wise. 14 markers is very low number. This should be discussed in conclusion as limitation of the research.

Moderate editing of English language required
Round 2
Reviewer 2 Report
Manuscript has been improved largely by the authors
In the revised PDF (V-2) file, any addition, deletion, changes are not clearly detected. Like, Fig. 1. Difficult to find out which one to accept.
Fig. 2: There are 3 figs. deletion not clear.
On the other hand: even though authors mentioned “Response 8: The information on K = 4 was removed from the entire document” however, nothing was changed in the revised doc. L: 109-128. If K-4 remains, please make clear by description in the text
Minor editing of English language required
